# Data Fusion of Fourier Transform Mid-Infrared (MIR) and Near-Infrared (NIR) Spectroscopies to Identify Geographical Origin of Wild *Paris*
*polyphylla* var. *yunnanensis*

**DOI:** 10.3390/molecules24142559

**Published:** 2019-07-13

**Authors:** Yi-Fei Pei, Zhi-Tian Zuo, Qing-Zhi Zhang, Yuan-Zhong Wang

**Affiliations:** 1Institute of Medicinal Plants, Yunnan Academy of Agricultural Sciences, Kunming 650200, China; 2College of Traditional Chinese Medicine, Yunnan University of Chinese Medicine, Kunming 650500, China

**Keywords:** origin traceability, data fusion, *Paris polyphylla* var. *yunnanensis*, Fourier transform mid-infrared spectroscopy, near-infrared spectroscopy

## Abstract

Origin traceability is important for controlling the effect of Chinese medicinal materials and Chinese patent medicines. *Paris polyphylla* var. *yunnanensis* is widely distributed and well-known all over the world. In our study, two spectroscopic techniques (Fourier transform mid-infrared (FT-MIR) and near-infrared (NIR)) were applied for the geographical origin traceability of 196 wild *P. yunnanensis* samples combined with low-, mid-, and high-level data fusion strategies. Partial least squares discriminant analysis (PLS-DA) and random forest (RF) were used to establish classification models. Feature variables extraction (principal component analysis—PCA) and important variables selection models (recursive feature elimination and Boruta) were applied for geographical origin traceability, while the classification ability of models with the former model is better than with the latter. FT-MIR spectra are considered to contribute more than NIR spectra. Besides, the result of high-level data fusion based on principal components (PCs) feature variables extraction is satisfactory with an accuracy of 100%. Hence, data fusion of FT-MIR and NIR signals can effectively identify the geographical origin of wild *P. yunnanensis*.

## 1. Introduction

The rhizome of *Paris polyphylla* var. *yunnanensis* (Franch.) Hand. -Mazz (*P. yunnanensis*) and *P. polyphylla* Smith var. *chinensis* (Franch.) Hara (*P. chinensis*), named as “chonglou” in Chinese, is a renowned and traditional herb with a history of thousands of years in China and plants belong to *Paris* genus, Liliaceae family. As an ancient history ethnobotanical medicinal plant, it is used to treat snake bite and insect sting, innominate toxin swelling, and a variety of inflammatory and traumatic in the folk in China. Various phytochemical researches have demonstrated that steroidal saponin, phytosterol, molting hormone, flavone, and pentacyclic triterpene are the major chemical components in *Paris* [1,2]. Additionally, *Paris* is considered to have anti-bacterial, anti-myocardial ischemia, anti-tumor, analgesia, immune-regulation according to numerous pharmacological studies [3,4,5,6,7]. As an important and precious medicinal plant, the raw material plants of *P. yunnanensis* are widely spread over southern China, especially in Yunnan Province [8]. Our previous studies have shown that there are significant differences in the content of wild *P. yunnanensis* samples from different geographical sources, and the saponin content in southern Yunnan is relatively higher than other regions [9,10]. Hence, it is crucial to the traceable geographical origin of wild *P. yunnanensis* samples to ensure effective medicinal values, which helps to ensure the effectiveness of the medication.

Some techniques have been applied to identify the authenticity and quality of various herbal medicines, including Fourier transform mid-infrared (FT-MIR), near-infrared (NIR), ultraviolet-visible (UV-Vis), Raman, liquid chromatography-mass spectrometry (LC-MS), high performance liquid chromatography (HPLC), etc. [11,12,13,14,15,16]. In recent years, several researches in classification of *P. yunnanensis* have widely used chemometrics models combined with various analytical techniques, including partial least squares discriminant analysis (PLS-DA), principal component analysis (PCA), hierarchical cluster analysis (HCA), random forest (RF), support vector machine (SVM), etc. [17,18,19,20,21]. Among them, spectroscopic techniques are fast, lossless, and efficient for the analysis of herbal medicines. Besides, quality of *P. yunnanensis* is difficult to identify by one or several chemical components due to the synergistic effect of TCMs, while the integral chemical information of medicinal plants can be provided by chromatographic or spectral fingerprints. For example, Yang et al. applied PCA and cluster analysis combined with FT-MIR to classify the small quantity wild and cultivated *P. yunnanensis* samples [21]. PLS-DA and RF models combined with FT-MIR have successfully traced the cultivated *P. yunnanensis* samples from Yunnan Province with different cultivation years [17]. Besides, our previous study has shown that the PLS-DA model combined with different parts (rhizomes and leaves) FT-MIR information can effectively distinguish the samples of cultivated *P. yunnanensis* collected in different cities of Yunnan Province [19].

Only a kind of chemical profile was obtained by the single technique, the relative complete chemical information would be provided by multiple platforms. The data fusion strategy contains low-, middle- and high-level, which can effectively fuse the chemical information obtained by different platforms of samples into one dataset to identification and classification researches [22]. As a case study, Li et al. found that the discriminant model established by FT-MIR and NIR spectral data combined with high-level data fusion strategy can effectively identify *Panax notoginseng* from different cultivation regions [23]. Wu et al. demonstrated that FT-MIR combined with UV-Vis by data fusion strategy could obtain a reliable and good result to trace the geographical origins of wild *P. yunnanensis* samples [24]. Hence, it is vital to effectively combine multiple techniques datasets of *P. yunnanensis* to obtain excellently chemical information to identification analysis and the effective results.

In this study, to obtain further realization of the similarities and differences among wild *P. yunnanensis* samples from central, western, northwest, southeast, and southwest Yunnan, we studied the collected samples using two spectroscopic techniques (FT-MIR and NIR), and fused with data fusion strategy (low-, mid- and high-level) combined with chemometrics including PCA, PLS-DA, and RF. The important variables regions among each classification models and the fast-quality assessment effects for geographical origins of *P. yunnanensis* were compared. Additionally, the chemical fingerprint of wild *P. yunnanensis* samples from different areas at Yunnan Province for FT-MIR and NIR spectra were analyzed. The results of our study may provide some basis for comprehensive utilization of *P. yunnanensis* resources.

## 2. Results and Discussion

### 2.1. Macroscopic Chemistry Components in IR Spectra

Averaged raw FT-MIR and NIR spectra of wild *P. yunnanensis* samples from central, northwest, western, southeast, and southwest Yunnan Province are shown in Figure 1. Twenty-five major common peaks were contained in FT-MIR spectra from these five regions, as shown in Figure 1a. The 4000 to 3700 cm^−1^ and 2620 to 1800 cm^−1^ absorptions were the FT-MIR spectral baseline area and diamond crystal spectral region, respectively, which areas provide invalid spectral information for this study [25]. Besides, regions of 3700 to 2620 cm^−1^ and 1800 to 1300 cm^−1^ were defined as characteristic areas in our study, which mainly contained C═O, C═C, and C–H stretching vibration as well as C–H bending vibration mode [21,26]. In addition, the region of 1300 to 650 cm^−1^ was the fingerprint region, which greatly contained C–O stretching vibration, C–C stretching vibration, C–OH bending vibration mode, as well as sugar skeleton vibration [27,28]. For all above these useful regions, FT-MIR spectra can be divided into five distinct ranges, including 3700 to 2000, 1800 to 1500, 1500 to 1200, 1200 to 900, and 900 to 650 cm^−1^ [29]. In the region of 3700 to 2000 cm^−1^, the broad absorption band in the range 3700–3000 cm^−1^ corresponds to the stretching vibrations of free hydroxyl groups ν(OH) and the groups involved in intra- and intermolecular hydrogen bonds [29]. Absorption at the peaks of 2928 and 2852 cm^−1^ were assigned to normal vibration mode such as the CH_3_ asymmetric normal vibration mode at 2960 to 2920 cm^−1^, CH_2_ asymmetric normal vibration mode at 2930 to 2900 cm^−1^, CH_3_ symmetric normal vibration mode at 2900–2880 cm^−1^, and CH_2_ symmetric normal vibration mode at 2860 to 2850 cm^−1^ [29]. Additionally, the region two to the region four were useful to deconvolute the bands into Lorentzian components, and the detailed information can be observed in Table 1. Moreover, amide I band is observed in the region of 1800 to 1500 cm^−1^ too, which corresponds to the C═C stretching mode of fatty acids and flavonoids [29]. 

Nine major common peaks are obtained in average NIR spectra from five geographical origins, as shown in Figure 1b. The bands in the region of 9000 to 4500 cm^−1^ be associated with the first or second overtones [30]. The peaks in the region 4500 to 4000 cm^−1^ are so narrow that it was difficult to provide detailed information [31]. Besides, the peaks at 5169, 3382, 3334, and 1653 cm^−1^ were considered, which also may correspond to hydrogen bond stretching and scissoring vibration mode attributed to water molecules [29]. The fingerprint of FT-MIR and NIR spectra characteristics of *P. yunnanensis* from different geographical origins were similar, as shown in Figure 1, which indicated similar chemical composition among these samples. Additionally, detailed peak positions and assignments were applied in Table 1. 

### 2.2. Single Block Models

Raw FT-MIR and NIR spectra were pretreated by standard normal variate (SNV), first derivative (FD), second derivative (SD), SNV-FD, and SNV-SD, and parameters of these pretreatment algorithms are applied in Appendix A, including parameters of cumulative interpretation ability (R^2^), cumulative prediction ability (Q^2^), the root mean square error of estimation (RMSEE), the root mean square error of cross-validation (RMSECV), and accuracy. For the FT-MIR spectra dataset, the worst classification ability was by FD algorithms, which also had accuracy worse than the raw dataset. But for the NIR spectra dataset, the SNV pretreatment algorithm was the worst preprocessing algorithm. However, SD was the best preprocessing algorithm, both for FT-MIR and NIR, whereby the accuracy even reached 100%. Among all preprocessing algorithms, the best pretreatment algorithm (SD) for each kind of spectroscopy should be selected and used to establish geographical classification models.

PLS-DA and RF classification models were established on FT-MIR and NIR SD datasets, respectively. The efficiency for each class and accuracy of the calibration set and validation set of these geographical origin models are shown in Table 2. The parameter of the root means square error of prediction (RMSEP) was one important parameter for evaluating model classification ability. The values of RMSEP were 0.203 and 0.236 of PLS-DA based on FT-MIR and NIR spectra datasets, respectively. With the higher accuracy and lower RMSEP, the PLS-DA model effect of using FT-MIR data to classify geographical origins was better than that of NIR.

The permutation test can be used to determine whether the established PLS-DA model is at risk of overfitting [32,33]. The intercepts generated by 200 random permutation tests for the FT-MIR PLS-DA model permutation test with the selected-class 1 variables were R^2^ = 0.395 and Q^2^ = −0.861, the values of R^2^ and Q^2^ of permutation tests with the remaining four categories variables are shown in Appendix A. All these results showed that this model was robust without overfitting. Additionally, the PLS-DA models involved in this paper were subjected to permutation tests and there was no overfitting.

For the RF model based on the FT-MIR spectra data matrix, the initial number of trees (*n*_tree_) and number of variables (*m*_try_) were set as 2000 and the square root of the number of variables. The optimal value of n_tree_ was selected based on the lowest total out-of-bag (OOB) classification error value, meanwhile assured of the lower OOB classification error values of the most classes. Besides, the optimal *n*_tree_ should be selected from the smooth region of the curve when the minimum OOB value was obtained in multiple regions. The optimal *m*_try_ was selected according to the lowest OOB classification error value. As shown in Figure 2a,b, the most suitable values 1780 and 33 were selected as the best *n*_tree_ and *m*_try_, respectively, for the RF model based on the FT-MIR spectra data matrix. Based on the same principle, the optimal values 434 and 39 were the best *n*_tree_ and *m*_try_, respectively, for the RF model based on the NIR spectra dataset, as shown in Figure 2c,d.

### 2.3. Important Variable Datasets Selected for Mid-Level Data Fusion

Mid-level on principal components (Mid-level-PCs), mid-level on recursive feature elimination (Mid-level-RFE) and mid-level on Boruta (Mid-level-Bo) dataset matrixes needed to be established to complete mid-level data fusion. The mid-level-RFE dataset was established as follows: 

RF models were established on FT-MIR and NIR spectra datasets of wild *P. yunnanensis* samples. For the two RF models, the initial n_tree_ for both was defined as 2000, and the initial m_try_ was set as 33 for the FT-MIR dataset and 39 for the NIR dataset. A total of 1033 trees and 529 trees were the optimal values for *n*_tree_ of FT-MIR and NIR datasets, respectively, which are shown in Appendix A. Based on the optimal *n*_tree_, the number of m_try_ was calculated to be 36 and 32 for FT-MIR and NIR spectra data matrixes, respectively. Next, the optimal *n*_tree_ and *m*_try_ were used to further obtain the importance of each variable of individual spectra matrix. All variables of FT-MIR and NIR datasets were arranged from small importance to large importance, respectively. The 10-fold cross-validation error rates of the RF model, based on FT-MIR and NIR data matrixes, are shown in Figure 3a,b. It was reduced sequentially by five variables for each step for the sorted important variables of FT-MIR and NIR spectra data matrixes, respectively. For the FT-MIR dataset (Figure 3a), all variables were divided into region 1, region 2, and region 3, which represent irrelevant variables, interference variables, and important variables, respectively [23]. Among them, region 3 remained for further research. In other words, 45 of the most important variables of the FT-MIR dataset were selected to prepare for mid-level data fusion. However, all variables of the NIR data matrix were divided into region 1 and region 2 (Figure 3b), which represent irrelevant variables and important variables. Variables of region 1 were excluded and the other 145 NIR variables of region 2 were used to fuse with important variables of the FT-MIR dataset to establish Mid-level-RFE models. In other words, these two block datasets straightforwardly concatenated and reconstituted the independent data matrix named Mid-level-RFE.

Basing on the optimal *n*_tree_ and *m*_try_, important (confirmed and tentative) variables were calculated to be 304 and 343 for NIR and FT-MIR spectra datasets, respectively. These two block datasets (important variables) reconstituted the independent data matrix named Mid-level-Bo. 

Similarly, two block datasets of 25 PCs NIR variables and 17 PCs FT-MIR variables straightforwardly concatenated and reconstituted the independent data matrix named Mid-level-PCs. 

Besides, the selected variables for establishing mid-level data fusion classification models using RFE and Bo algorithms are shown in Figure 4. The important (confirmed) and tentative variables are represented by blue lines and yellow lines, respectively. 

### 2.4. Important Variables Datasets Selected for High-Level Data Fusion

High-level data fusion uses the same PCs as mid-level data fusion as the PCs were selected from the unsupervised PCA model. Hence, 17 PCs of the FT-MIR spectra dataset (FT-MIR-PCs) and 25 PCs of the NIR spectra data matrix (NIR-PCs) were used to establish PLS-DA and RF models, respectively. For FT-MIR-PCs and NIR-PCs RF models, the two parameters of *n*_tree_ and *m*_try_ needed to be optimized first. As shown in Appendix A, the optimal values 535 and 1030 trees were selected and; furthermore, calculated the suitable *m*_try_ to be 4 and 3 for FT-MIR-PCs and NIR-PCs RF models, respectively. Then, final RF models were established on the optimal parameters and the most important variables. Besides, vote results of validation sets and calibration sets of PLS-DA and RF models based on the FT-MIR-PCs and NIR-PCs datasets were obtained as shown in Appendix A. 

For the RF model based on the FT-MIR and NIR spectra datasets, the initial *n*_tree_ was set as 2000, and the initial *m*_try_ were set as 33 (FT-MIR) and 39 (NIR), respectively. The 1780 trees and 434 trees are the optimal values for *n*_tree_ of FT-MIR and NIR datasets (Figure 2). The numbers of *m*_try_ were defined to be 33 and 39 for FT-MIR and NIR spectra data matrixes, respectively, based on the optimal *n*_tree_. Furthermore, all variables of FT-MIR and NIR datasets were sorted, respectively. The 10-fold cross validation error rates of the RF model, based on the FT-MIR and NIR data matrixes are shown in Appendix A. For the FT-MIR dataset (Appendix A), all variables were divided into three regions. Among them, 80 of the most important variables (region 3) of the FT-MIR dataset were selected to establish FT-MIR-RFE PLS-DA and RF models. All variables of the NIR data matrix were divided into two regions (Appendix A) and 200 NIR variables of region 2 were used to establish NIR-RFE PLS-DA and RF models. PLS-DA and RF models were based on two important variables datasets (FT-MIR-RFE and NIR-RFE), respectively. For FT-MIR-RFE and NIR-RFE RF models, 293 and 1459, respectively, were selected as the suitable trees, as shown in Appendix A. Furthermore, the optimal *m*_try_ values were calculated to be 8 and 14, respectively. Based on the optimal parameters, FT-MIR-RFE and NIR-RFE datasets were used to establish the RF model. Similarly, vote results of validation sets and calibration sets of PLS-DA and RF models based on FT-MIR-RFE and NIR-RFE data matrixes could be obtained, respectively, as shown in Appendix A.

The FT-MIR-Bo and NIR-Bo datasets were obtained by selecting important variables using the Bo algorithm with FT-MIR and NIR spectra datasets. These two data matrixes were used to establish PLS-DA and RF models, respectively. As shown in Appendix A, 1143 and 875 trees were selected to be the optimal n_tree_ for FT-MIR-Bo and NIR-Bo datasets, respectively. Besides, the suitable m_try_ values of the two datasets were calculated to be 12 and 20, respectively. Similarly, vote results of validation sets and calibration sets of two models were obtained as shown in Appendix A.

Like mid-level data fusion, the comparison of selected variables for establishing high-level data fusion classification models using two algorithms is shown in Appendix A. In addition, it was found that both the fingerprint region and characteristic region variables contribute to the classification of *P. yunnanensis* from different origins.

The results of PLS-DA and RF models based on RFE, Bo, and PCs selection algorithms are shown in Table 2. For FT-MIR datasets containing three-variable selection algorithms, the validation set classification results of the PLS-DA and RF models were similar and slightly lower than that obtained by models based on the raw FT-MIR data matrix. Among these models, the RF model using the PCs selection algorithm showed the best ability. Besides, the classification ability of the PLS-DA and RF models based on NIR (RFE) and NIR (Bo) data matrixes were significantly lower than that based on the NIR (PCs) dataset. However, the accuracy for the validation sets of the PLS-DA and RF models based on the NIR (PCs) data matrix were both higher than that based on the FT-MIR (PCs) dataset, which is contrary to the results based on original FT-MIR and NIR datasets. Additionally, it is not hard to see that the classification ability of the PLS-DA and RF models based on the RFE algorithm were like the Bo algorithm, no matter whether based on the FT-MIR or the NIR spectral data. Distribution of their important variables was almost the same (Appendix A), which may be the reason for the similar classification results. Bo variables selection algorithm would be the preferred one than RFE algorithm because it used lesser operation time. 

### 2.5. Low-Level Data Fusion Models

In this case, the low-level data matrix (196 × 2695) was used to establish PLS-DA and RF models. For the low-level data fusion RF model, 2000 and 51were set as original *n*_tree_ and *m*_try_ values, respectively. As shown in Appendix A, 802 (*n*_tree_) and 51 (*m*_try_) were selected as the optimal parameters to establish the final low-level data fusion RF model. The results of calibration and validation sets for the two kinds of models are reported in Table 3. Although the accuracy of the calibration sets of the two models was quite different, the accuracy of the validation sets was only 3% different. Compared with the accuracy of individual data source models, the accuracy of low-level data fusion by both models was similar to that of the FT-MIR dataset, which was higher than that of the NIR data matrix. Besides, the samples of the second class (Northwest Yunnan) were correctly predicted by both low-level data fusion models. The low-level data fusion strategy enhanced (to 100%) the accuracy of samples from Northwest Yunnan. 

Variables whose VIP (Variable importance in the projection) score was greater than 1 were selected as important variables to establish PLS-DA and RF models. The selected variables distribution is shown in Appendix A and results of classification models are displayed in Table 3. The important variables were lesser dispersed in regions 3580 to 3530 cm^−1^, 3460 to 3150 cm^−1^, 3000 to 2950 cm^−1^, 2750 to 2720 cm^−1^, 1400 to 1200 cm^−1^, and 1000 to 800 cm^−1^ of FT-MIR spectra and regions 6540 to 6200 cm^−1^ and 5070 to 4000 cm^−1^ of NIR spectra. Many interference and unrelated variables were eliminated by selecting VIP values. The efficiency and accuracy of each class in the low-level (VIP) data fusion model was unchanged and even enhanced. 

### 2.6. Mid-Level Data Fusion Models

In our study, Mid-level-PCs, Mid-level-RFE, and Mid-level-Bo three kinds of data matrixes were used to establish PLS-DA and RF classification models to prepare for mid-level data fusion. As shown in Appendix A, 427 (*n*_tree_) and 6 (*m*_try_) were selected as optimal parameters to establish the final mid-level-PCs RF model and obtained a validation set accuracy of 94.12%. The total accuracy of the validation set of the PLS-DA model was 98.53% (Table 3).

Similarly, 2000 and 13 were set as raw n_tree_ and m_try_ values for the mid-level-RFE RF model. According to the principle of parameter optimization of the RF model, the lowest values 262 (*n*_tree_) and 13 (*m*_try_) were defined as the suitable parameters to establish the RF model, as shown in Appendix A. However, the validation set accuracy of the mid-level-RFE was only 69.12%, which was lesser than that of each individual spectral RF model. Besides, the PLS-DA model based on the mid-level-RFE had a worse classification ability with an accuracy of only 55.88% (Table 3).

The numbers of 2000 and 25 were set as raw *n*_tree_ and *m*_try_ values, respectively, for the mid-level-Bo RF model. As shown in Appendix A, the suitable parameters were calculated to be 800 and 15, to establish the mid-level-Bo RF model, and the obtained validation set accuracy was 95.59%. The difference of accuracy for both the PLS-DA model and the RF model based on mid-level-RFE and mid-level-Bo was about 30%. By comparing the regions of important variables between mid-level-RFE and mid-level-Bo data matrixes (Figure 4), we could find that the number of important FT-MIR variables for the RFE variable selection algorithm was less than that of the Bo variable selection algorithm, and there was little difference in the important variables of the NIR spectroscopy selected by the two algorithms. Hence, we can infer that the RFE selection algorithm had excluded some of the important variables that may be enhancing the accuracy of classification models. In addition, we can also extrapolate that the FT-MIR dataset was more important for geographical origin classification of wild *P. yunnanensis* samples than the NIR data matrix, which provides more effective information.

### 2.7. High-Level Data Fusion Models

For high-level data fusion, four fuzzy aggregation operators were chosen as the voting rule for the voting decision, including minimum, maximum, product, and average [23]. The category that has the maximum value in each operator is considered to be the selected class. It is worth mentioning that when the difference between the maximum value and the second largest value is less than 0.01, both values are considered maximum. Three kinds of vote results would be obtained by high-level data fusion including correct, false, and multiple discriminated. As shown in Appendix A, the true Class of sample NO. 4 belongs to Class 1 and the four fuzzy aggregation operators were fully accorded with the true Class. For example, for No. 31 the true category is Class 1, while three voting results are distinguished into Class 3 and one voting result is defined as Class 5. The high-level data fusion voting results of this sample is Class 3. Besides, NO. 114 was voted into Class 2 and Class 3 by FT-MIR-PCs and NIR-PCs RF models. This sample truly belongs to Class 3, while the final data fusion voting result is distinguished into Class 2. Besides, sample 6 truly belongs to Class 1 while pertained to Class 1 and Class 4 by voting, which was multiple discriminated. Although multivariate discrimination does not affect the accuracy of the model, it influences efficiency values of each Class. This explains that the accuracy of the validation set of RF model is 1, while the efficiency of Class 1 and Class 4 does not reach 1 (Appendix A).

As shown in Table 3, the accuracy of the validation set of the High-level-PCs RF model was reached at 100% and that of the High-level-PCs PLS-DA model was 98.53%. The high-level data fusion classification ability based on the PCs selection variables algorithm was better than that based on RFE and Bo algorithms. Besides, there was little difference between the PLS-DA model and the RF model in identifying the origin of *P. yunnanensis* samples based on the same data set.

## 3. Materials and Methods

### 3.1. Samples Preparation

The 196 rhizomes of wild *P. yunnanensis* samples were obtained from five different origins at central, western, northwest, southeast, and southwest areas of Yunnan Province, as shown in Figure 5. The detail collection information is shown in Appendix A. All wild samples were identified as *P. polyphylla* var. *yunnanensis* (Franch.) Hand. -Mazz. by Professor Hang Jin (Institute of Medicinal Plants, Yunnan Academy of Agricultural Sciences, Kunming, China). All rhizome samples were washed with tap water and were dried in a drying oven at 50 °C, then sifted through 100 mesh sieves. Additionally, all samples were preserved in polyethylene zip-lock bags and kept in a dark and dry environment for further analysis. 

### 3.2. Fourier Transform Mid-Infrared Spectroscopy (FT-MIR)

FT-MIR spectra were collected with an FTIR spectrometer equipped with a deuterated triglycine sulfate (DTGS) detector and a ZnSe ATR (attenuated total reflection) accessory (Perkin Elmer, Norwalk, CT, USA). All spectra recorded ranges of 4000 to 650 cm^−1^ with 4 cm^−1^ resolution, and 16 scans were averaged. Three analytical replicates of FT-MIR spectral data of all wild *P. yunnanensis* samples were obtained.

### 3.3. Near-Infrared Spectroscopy (NIR)

NIR analysis was conducted with an Antaris II spectrometer (Thermo Fisher Scientific, Madison, WI, USA) equipped, combined with a diffuse reflection module. All spectra recorded ranges of 10,000 to 4000 cm^−1^ with a spectral resolution of 4 cm^−1^, and 16 scans were averaged for wild samples. Three scans were repeated for all wild samples.

### 3.4. Spectral Data Analysis and Software

FT-MIR spectra were converted from transmittance to absorbance and the advanced ATR correction was completed by OMNIC 9.7.7 software (Thermo Fisher Scientific, Madison, WI, USA). The spectral linear relation was greatly disturbed by high-frequency random noise, the interference of light scattering, baseline drift, etc. [34]. Hence, FT-MIR and NIR spectra were processed using SNV, FD, SD, and their combination (SNV-FD and SNV-SD), to decrease a part of the irrelevant interferences [10,35,36]. All these pretreatment procedures were performed by SIMAC-P^+^ (Version 13.0, Umetrics, Umeå, Sweden). Additionally, the spectral regions of 4000 to 3700 cm^−1^ and 2620 to 1800 cm^−1^ were excluded for all FT-MIR spectra before establishing classification models due to a mass of interference information. Hence, the regions of 3700 to 2620 cm^−1^ and 1800 to 650 cm^−1^ formed a data matrix for constructing classification models.

All samples for each class were separated into calibration sets and validation sets as a rate of 2 to 1 with Kennard-Stone algorithm using MATLAB (Version R2017a, Mathworks, Natick, MA, USA) [37,38]. In other words, the number of calibration sets (128 samples) in one to five categories was 26, 26, 24, 26, and 26, respectively, and the number of validation sets (68 samples) in one to five classes was 14, 14, 12, 14, and 14, respectively. The preprocessing algorithms were estimated by parameters of R^2^, Q^2^, RMSEE, RMSECV, and accuracy of the calibration set [19,39]. The better pretreatment algorithm required higher values of R^2^, Q^2^, and accuracy as well as lower values of RMSEE and RMSECV. Hence, the best preprocessing algorithm would be selected for identification analysis to establish PLS-DA and RF classification models. 

The groundwork of PLS-DA is the PLS algorithm and it belongs to the binary classification algorithm from 0 to 1, which has been widely applied to resolve the classification problems for geographical origins, growth years, and others [40]. For each sample, the probability of being assigned to each class could be obtained, and the category with the highest probability was seen as the category of this sample. For the validation samples, RF is based on the assembly classification or regression trees algorithm and has a better ability to handle the nonlinear and high-order interaction effects data matrixes [41]. Both of these kinds of class-modeling methods belong to supervised pattern recognition and require a calibration set for each class in order to establish an individual model to explore their similarities between samples from the one class and the differences among all classes. Besides, the validation sets were used to validate the identification ability of supervised models. In our study, all PLS-DA were completed by SIMAC-P^+^ (Version 13.0, Umetrics, Umeå, Sweden) and RF were established by RStudio (version 3.5.2, Boston, MA, USA). The operation of RF was roughly divided into the following three steps: Firstly, 2000 and the square root of the number of variables were set as the initial values of n_tree_ and m_try_, respectively. Secondly, these two parameters were optimized according to the lowest OOB classification error values. Thirdly, the RF model was established with the selected optimal values of *n*_tree_ and *m*_try_.

The RMSEP and accuracy of the validation set were the two parameters used to estimate the classification ability of PLS-DA. The lower values of RMSEP shows the better prediction ability of PLS-DA models. Besides, for the PLS-DA and RF models established by the best preprocessing algorithm, indices of true positives (TP), true negatives (TN), false positives (FP), and false negatives (FN) were calculated for each class. The sensitivity (SEN) and specificity (SPE) were obtained by the above indices for each class. The efficiency of values was calculated by the geometric mean of SEN and SPE to evaluate the effectiveness of each class of PLS-DA and RF models. All formulas for the above parameters were as follow:(1)$$\mathrm{SEN}=\frac{\mathrm{TP}}{\mathrm{TP}+\mathrm{FN}}$$
(2)$$\mathrm{SPE}=\frac{\mathrm{TN}}{\mathrm{TN}+\mathrm{FP}}$$
(3)$$\mathrm{efficiency}=\sqrt{\mathrm{SEN}\times \mathrm{SPE}}$$

The map of sample collection information in our study was obtained by Arc Map (version 10), and all figures were drawn by Origin (version 2018, OriginLab Corporation, Northampton, MA, USA) and Adobe Photoshop CC (version 2019, Adobe Systems Incorporated, San Jose, CA, USA). 

### 3.5. Data Fusion Strategy

Data fusion strategy was applied in this study, as the comprehensive information of *P. yunnanensis* samples were unable to be provided by individual data sources. To compare and select the best data fusion strategy to trace geographical origins, the low-, mid-, and high- level data fusion strategies and three algorithms for variable selection were considered. The best preprocessing FT-MIR and NIR datasets were used to finish data fusion approaches. The schemes for these three strategies combined with two kinds of spectral signals are shown in Figure 6. 

In low-level data fusion strategy, the FT-MIR and NIR spectral signals are straightforwardly concatenated and reconstitute an independent data matrix. This new dataset (low-level data matrix) was equal to 196 rows and 2695 columns, namely 196 samples and 2695 spectral variables (= 1545 NIR variables + 1150 FT-MIR variables). Finally, the low-level dataset was used to establish the PLS-DA and RF models.

Mid-level data fusion strategy, namely feature-level data fusion, was made up of feature important variables from single data sources including FT-MIR and NIR spectra. PLS-DA and RF classification models were established by new data matrixes, which were formed by concatenating the feature important variables from FT-MIR and NIR by different variable selection algorithms. In this case, important variables were selected based on each Y parameter (spectral datasets of total samples). More in detail, three different variable selection algorithms were as follows:Mid-level-PCs consisted of principal components, which were selected by PCA of FT-MIR and NIR spectral datasets, respectively. PCs were selected based on values of eigenvalue greater than 1. Hence, the mid-level-PCs data matrix was obtained with 196 rows and 42 columns, namely 196 samples and 42 PCs variables (= 25 NIR PCs variables + 17 PCs FT-MIR variables).Mid-level-RFE, which consisted of merging together the important variables of FT-MIR and NIR spectral datasets, was selected by the recursive feature elimination algorithm based on the RF model. The mid-level-REF dataset size was equal to 196 rows and 190 columns (= 145 NIR REF variables + 45 FT-MIR REF variables).Mid-level-Bo, which consisted of merging together the important (confirmed and tentative) variables of FT-MIR and NIR spectral datasets, was selected by the Boruta algorithm based on the RF model. The mid-level-Bo data matrix consisted of 196 rows and 647 columns (= 153 NIR Bo confirmed variables + 151 NIR Bo tentative variables + 207 FT-MIR Bo confirmed variables + 136 FT-MIR Bo tentative variables).

High-level data fusion strategy, namely decision-data fusion, fused the vote results from the models of the FT-MIR and NIR datasets. Additionally, individual spectral matrices were formed by feature or important variables using variable selection algorithms (PCs, RFE, and Bo) before establishing models. In our study, important variables were selected by RFE and Bo for the high-level data fusion strategy based on the Y parameter of the calibration set, which was different from the mid-level data fusion variables selection method.

NIR-PCs data matrix obtained 196 rows and 25 columns (25 NIR PCs variables) and the FT-MIR-PCs data matrix obtained 196 rows and 17 columns (17 FT-MIR PCs variables).NIR-RFE data matrix obtained 196 rows and 200 columns (200 NIR RFE variables) and the FT-MIR-RFE data matrix obtained 196 rows and 80 columns (80 FT-MIR RFE variables).NIR-Bo data matrix obtained 196 rows and 183 columns (83 NIR Bo confirmed variables + 108 NIR Bo tentative variables) and the FT-MIR-Bo data matrix obtained 196 rows and 226 columns (117 FT-MIR Bo confirmed variables + 109 FT-MIR Bo tentative variables).

Besides, PCs were obtained using MATLAB (Version R2017a, Mathworks, Natick, MA, USA) and RFE and Bo were completed by RStudio (version 3.5.2).

## 4. Conclusions

In our study, the use of low-, mid-, and high-level data fusion strategies, combined with feature extraction and important variable selection algorithms, were researched to fuse the chemical information from FT-MIR and NIR spectroscopies for the identification and classification of geographical origins of wild *P. yunnanensis* samples. 

In fact, PCs was the feature extraction algorithm of three kinds of important variable selection algorithms, which obtained a better ability for establishing classification models no matter whether in mid- or high-level data fusion. Between the two important variable selection algorithms of RFE and Bo, the latter can obtain important variables that are similar, or more accurate, to the former and can complete the calculation in a shorter time.

Besides, the two kinds of IR spectroscopies bring complementary chemical information profiles about multiple geographical sources of *P. yunnanensis*. While FT-MIR provides chemical information among 4000 to 650 cm^−1^, NIR describes the chemical information from 10,000 to 4000 cm^−1^. The data fusion strategy improved the geographical traceability ability of models for *P. yunnanensis*, while FT-MIR spectra data provided more contributions than NIR spectra. Besides, thanks to the application of the high-level data fusion strategy, the identification effect based on the random forest model reached the best performance level. 

## Figures and Tables

**Figure 1 molecules-24-02559-f001:**
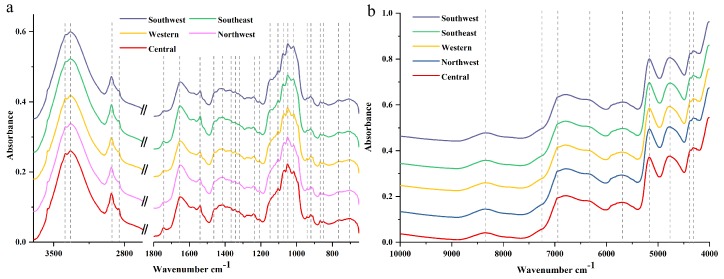
Averaged spectra of *P. yunnanensis* samples collected from five regions: (**a**) Fourier transform mid-infrared (FT-MIR) spectra; (**b**) near-infrared (NIR) spectra.

**Figure 2 molecules-24-02559-f002:**
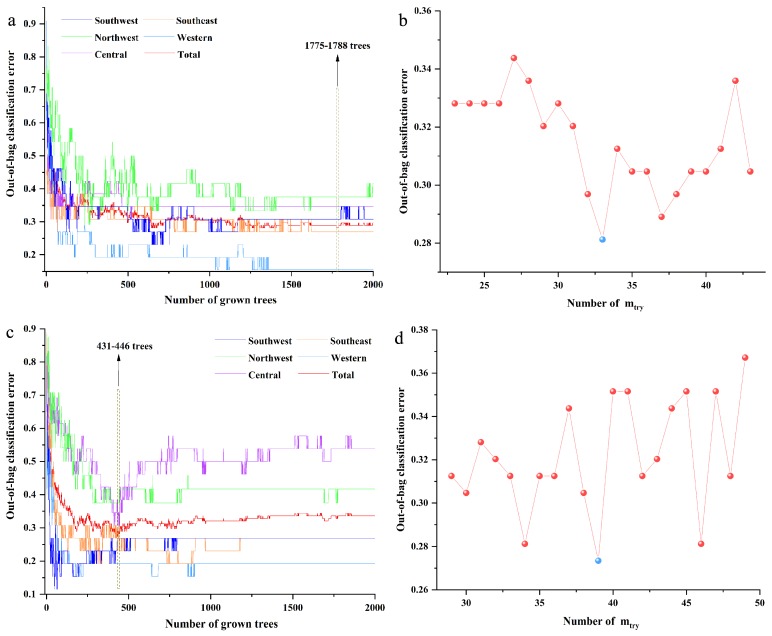
The parameter optimization of random forest models of independent decision making: (**a**) number of trees (*n*_tree_) of the FT-MIR dataset; (**b**) number of variables (*m*_try_) of the FT-MIR dataset; (**c**) *n*_tree_ of the NIR dataset; (**d**) *m*_try_ of the NIR dataset.

**Figure 3 molecules-24-02559-f003:**
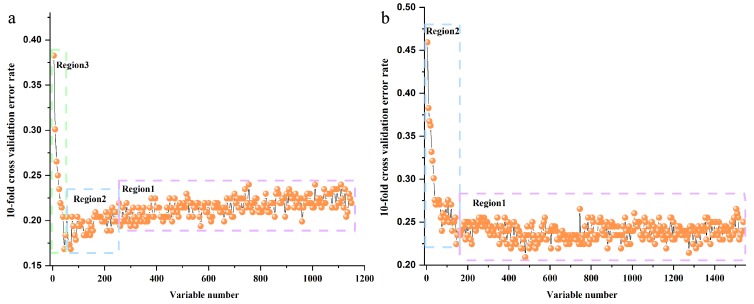
The 10-fold cross-validation error rates of the Random Forest (RF) model (sequentially reduced every five variables) based on total *P. yunnanensis* samples: (**a**) FT-MIR dataset; (**b**) NIR dataset.

**Figure 4 molecules-24-02559-f004:**
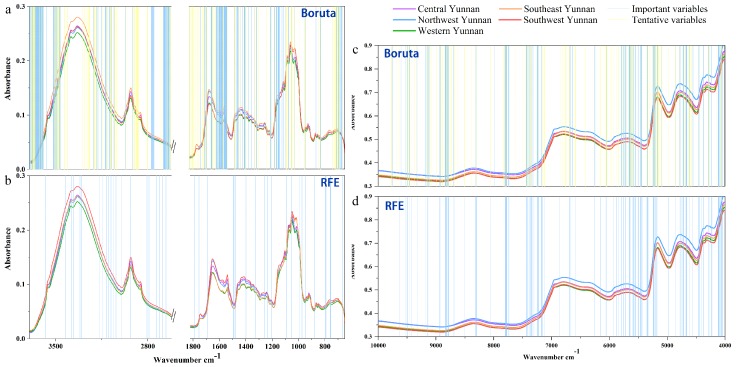
The important variables of Boruta algorithm and RFE algorithm of random forest models based on total *P.*
*yunnanensis* samples: (**a**,**b**) the FT-MIR dataset; (**c**,**d**) the NIR dataset. RFE: Recursive feature elimination.

**Figure 5 molecules-24-02559-f005:**
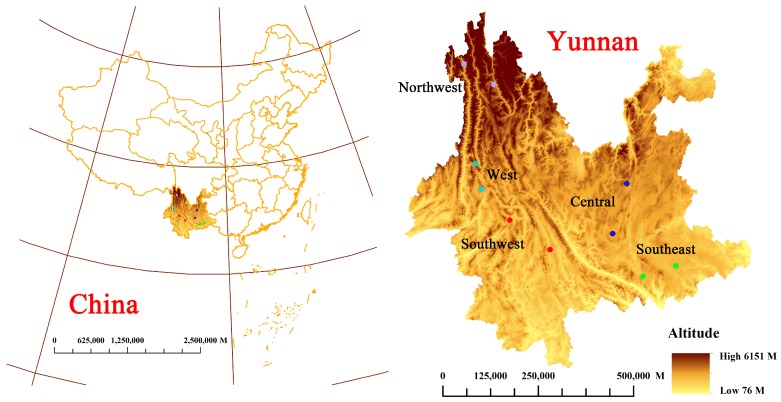
Location distribution of wild *P. yunnanensis* samples in central, western, northwest, southeast, and southwest areas, Yunnan Province.

**Figure 6 molecules-24-02559-f006:**
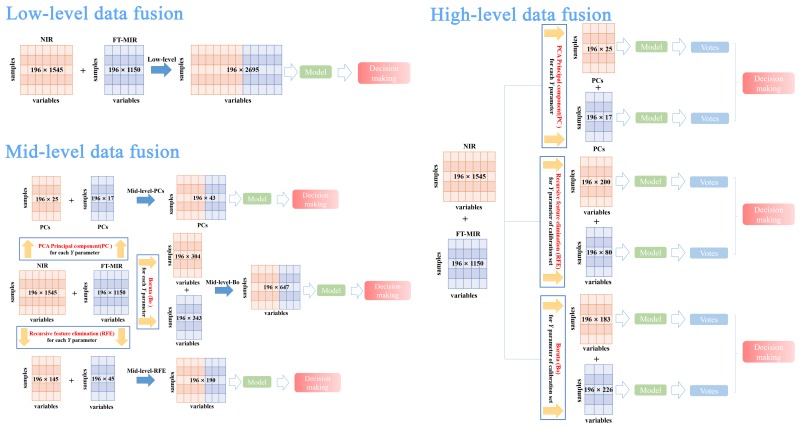
Scheme of the low-, mid-, and high-level data fusion approaches used to combine the FT-MIR signals and NIR signals.

**Table 1 molecules-24-02559-t001:** Peak assignments on the FT-MIR and NIR spectra of wild *P. yunnanensis*.

Spectral Type	Wavenumber (cm^−1^)	Base Group and Vibration Mode	Contribution
NIR	8347	C–H, N–H and O–H stretching vibration mode	CH_2_, saccharides, and glycosides
	7256	C–H stretching and deformation vibration mode	CH_2_
	6950	C–H, N–H and O–H stretching vibration mode	CH_2_, saccharides, and glycosides
	6324	C–H, N–H and O–H stretching vibration mode	CH_2_, saccharides, and glycosides
	5686	C–H, N–H and O–H stretching vibration mode	CH_2_, saccharides, and glycosides
	5169	C–H, N–H and O–H and hydrogen bond stretching vibration mode	CH_2_, saccharides, glycosides, and water molecule
FT-MIR	3382	O–H asymmetric and hydrogen bond stretching vibration mode	Saccharides, glycosides, and water molecule
	3334	O–H asymmetric and hydrogen bond stretching vibration mode	Saccharides, glycosides, and water molecule
	2930	C–H asymmetric stretching vibration mode	CH_2_ and CH_3_
	1743	C═O stretching vibration mode	Free carboxyl groups of pectins or/and fatty acids
	1653	asymmetric stretching vibrations of carboxyl groups participating in the hydrogen bonds and hydrogen bond scissoring vibration mode	Flavonoids, saccharides, steroid saponin, and water molecules
	1610	COO symmetric normal vibrations mode	The carboxyl group present in pectin
	1456	CH_3_ asymmetric deformation and CH_2_ scissoring vibration	CH_2_ and CH_3_
	1414	C–H symmetric bending vibration mode and OH–O in-plane bending mode	CH_2_
	1370	C–H symmetric deformation vibration mode	CH_3_
	1242	C–O stretching vibration mode	Saccharides and oils
	1150	C–C and C–O stretching and C–OH bending vibration mode	Saccharides and glycosides
	1078	C–C and C–O stretching and C–OH bending vibration mode	Saccharides and glycosides
	1020	C–C and C–O stretching and C–OH bending vibration mode	Saccharides and glycosides
	922	Sugar skeleton vibration mode	Saccharides

**Table 2 molecules-24-02559-t002:** The classification efficiency values and total accuracy of independent decision making with Partial least squares discriminant analysis (PLS-DA) and random forest (RF) models. RFE: Recursive feature elimination.

	Model	Calibration Set						Validation Set					
		Class1	Class2	Class3	Class4	Class5	Accuracy	Class1	Class2	Class3	Class4	Class5	Accuracy
FT-MIR	PLS-DA	0.961	1.000	0.995	0.981	0.990	97.66%	1.000	0.991	0.913	1.000	0.991	97.06%
	RF	0.772	0.888	0.801	0.829	0.790	71.88%	0.886	0.964	0.973	1.000	0.946	92.65%
NIR	PLS-DA	1.000	1.000	1.000	1.000	1.000	100%	0.870	0.964	0.940	0.936	0.964	89.71%
	RF	0.803	0.854	0.775	0.837	0.834	72.66%	0.813	0.917	0.491	0.923	0.955	76.47%
FT-MIR (RFE)	PLS-DA	0.911	0.990	0.881	0.961	0.975	91.41%	0.794	1.000	0.694	0.955	0.923	82.35%
	RF	0.947	0.951	0.853	0.876	0.942	86.72%	0.845	0.917	0.964	0.964	0.936	88.24%
FT-MIR (Bo)	PLS-DA	0.951	0.961	0.995	0.961	0.985	95.31%	0.886	0.991	0.905	0.964	0.962	91.18%
	RF	0.890	0.942	0.829	0.881	0.922	83.59%	0.886	0.964	0.973	1.000	0.946	92.65%
FT-MIR (PCs)	PLS-DA	0.906	0.911	0.868	0.927	0.863	83.59%	0.926	0.991	0.843	0.926	0.972	89.71%
	RF	0.780	0.922	0.730	0.764	0.772	68.75%	0.964	1.000	0.991	0.917	0.991	95.59%
NIR (RFE)	PLS-DA	0.807	0.922	0.926	0.902	0.966	85.16%	0.779	0.845	0.675	0.891	0.953	75%
	RF	0.733	0.888	0.791	0.888	0.942	77.34%	0.813	0.964	0.567	0.889	0.955	77.94%
NIR (Bo)	PLS-DA	0.807	0.922	0.858	0.906	0.947	82.81%	0.779	0.837	0.551	0.962	0.962	75%
	RF	0.729	0.878	0.764	0.893	0.922	75.78%	0.772	0.878	0.486	0.870	0.955	72.06%
NIR (PCs)	PLS-DA	0.860	0.937	0.974	0.915	0.990	89.84%	0.927	0.926	0.991	0.878	0.955	90%
	RF	0.745	0.922	0.881	0.881	0.951	81.25%	0.955	0.964	1.000	1.000	0.991	97.06%

Bo: Boruta, PCs: Principal components.

**Table 3 molecules-24-02559-t003:** The classification efficiency values and total accuracy of low-, mid-, and high-level data fusion strategies decision making with PLS-DA and RF models. RFE: Recursive feature elimination, Bo: Boruta, PCs: Principal components, VIP: Variable importance in the projection.

	Model	Calibration Set						Validation Set					
		Class1	Class2	Class3	Class4	Class5	Accuracy	Class1	Class2	Class3	Class4	Class5	Accuracy
Low-level	PLS-DA	1.000	1.000	1.000	1.000	1.000	100%	0.926	1.000	0.949	1.000	0.981	95.59%
	RF	0.872	0.885	0.858	0.897	0.932	82.81%	0.886	1.000	0.905	0.972	0.991	92.65%
Low-level (VIP)	PLS-DA	1.000	1.000	1.000	1.000	1.000	100%	0.926	1.000	0.991	1.000	0.991	97.06%
	RF	0.927	0.927	0.849	0.922	0.947	86.72%	0.926	1.000	0.991	1.000	0.991	97.06%
Mid-level (RFE)	PLS-DA	0.946	0.712	0.764	0.864	0.878	75%	0.705	0.794	0.000	0.727	0.798	55.88%
	RF	0.966	0.888	0.868	0.894	0.881	84.38%	0.764	0.861	0.491	0.900	0.854	69.12%
Mid-level (Bo)	PLS-DA	0.961	1.000	1.000	0.995	0.995	98.44%	0.926	1.000	0.991	0.991	1.000	97.06%
	RF	0.947	0.942	0.885	0.922	0.951	89.06%	0.926	1.000	0.949	0.991	0.991	95.59%
Mid-level (PCs)	PLS-DA	0.951	0.961	0.974	0.995	0.995	96.09%	1.000	1.000	0.957	0.991	1.000	98.53%
	RF	0.927	0.951	0.922	0.922	0.981	90.63%	0.886	1.000	0.991	0.981	0.955	94.12%
High-level (RFE)	PLS-DA	0.951	0.981	0.953	1.000	0.990	96.09%	0.870	1.000	0.802	0.991	0.981	89.71%
	RF	0.976	0.976	0.904	0.922	0.995	91.21%	0.926	1.000	0.991	1.000	0.991	97.06%
High-level (Bo)	PLS-DA	0.976	0.981	0.979	1.000	0.990	97.66%	0.926	0.991	0.850	1.000	0.981	92.65%
	RF	0.966	0.976	0.904	0.902	0.951	90.63%	0.917	0.964	0.850	0.972	1.000	91.18%
High-level (PCs)	PLS-DA	0.981	1.000	0.990	0.981	1.000	98.44%	0.964	1.000	1.000	0.991	1.000	98.53%
	RF	0.881	0.971	0.872	0.911	0.961	87.5%	0.966	1.000	1.000	0.991	1.000	100%

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
