# Peer review of "Data Fusion of Fourier Transform Mid-Infrared (MIR) and Near-Infrared (NIR) Spectroscopies to Identify Geographical Origin of Wild Paris polyphylla var. yunnanensis"

_molecules, 2019, doi:10.3390/molecules24142559_

Round 1

Reviewer 1 Report

General remarks:
a.    Please correct the English, vast fragments of the manuscript are difficult to understand.
b.    In my opinion this manuscript is too long. Authors devote to much place and effort to describe and discuss different models which are of similar quality. Some repetitions are also present. Please condense the text.
c.    I am not convinced that proposed procedures really improve classification. Accuracy, which is not the best measure, changes from 97-98% for ‘raw’ MIR data to a little bit more than 98% for ‘high-level’ PLS-DA model obtained for fused data. On the other hand for ‘low-level’ fused data its value is of the same order. Models with higher accuracy values for validation than for calibration samples, in my opinion, should be considered with care.

Other remarks:
a.    Authors use terms and expressions which are not commonly accepted or have other meaning than intended.
b.    Assignment proposed in Table 1 neglects water bands. What is the moisture content in the studied material?
c.    Some minor mistakes can be found in the reference section.
d.    Table 4 can be transferred to the supporting material section.

Author Response

Dear Reviewer,

Thank you for your letter and reviewers’ comments concerning about our manuscript entitled “Data fusion infrared signals to identify geographical origin of wild Paris polyphylla var. yunnanensis” (ID: molecules-517710). Those comments are valuable and very helpful for revising and improving our paper, as well as the important guiding significance to our researches. We have read comments carefully and have made corrections which we hope to meet with approval. Revised portion are marked in red font in the revised paper. The main corrections and the responds to the reviewer’s comments are displayed below:

Reviewer #1:

Questions:

General remarks:

a.               Please correct the English, vast fragments of the manuscript are difficult to understand.

Thanks for your advice. The style, spelling and grammar of this manuscript were carefully checked out and revised into the right form.

b.              In my opinion this manuscript is too long. Authors devote to much place and effort to describe and discuss different models which are of similar quality. Some repetitions are also present. Please condense the text.

Thanks for your advice. In this manuscript, we studied the collected samples by two spectroscopic techniques (FT-MIR and NIR) and fused with data fusion strategy (low-, mid- and high-level) combined with chemometrics including PCA, PLS-DA and RF. Besides, feature variables extraction (principal component analysis-PCA) and important variables selection models (recursive feature elimination and Boruta) were applied. Hence, it could require a more detailed description to express the meaning of these expression more clearly.

c.    I am not convinced that proposed procedures really improve classification. Accuracy, which is not the best measure, changes from 97-98% for ‘raw’ MIR data to a little bit more than 98% for ‘high-level’ PLS-DA model obtained for fused data. On the other hand for ‘low-level’ fused data its value is of the same order. Models with higher accuracy values for validation than for calibration samples, in my opinion, should be considered with care.

Thanks for your advice. The accuracy of validation set of PLS-DA and RF models basing on FT-MIR or NIR datasets were not reached to 100%, the accuracy of high-level data fusion has reached to 100% and samples only between two classes have been misclassified. In fact, the purpose of this study was to select the higher validation set accuracy of models. Hence, I think the conclusion of this manuscript is right.

Other remarks:

a.              Authors use terms and expressions which are not commonly accepted or have other meaning than intended.

Thanks for your advice. The terms and expressions of this manuscript were carefully checked out and revised into the right form.

b.             Assignment proposed in Table 1 neglects water bands. What is the moisture content in the studied material?

Thanks for your advice. I am sorry that the moisture content in this studied material were not measured. Besides, the more detailed information of spectral resolution has been added in manuscript.

c.              Some minor mistakes can be found in the reference section.

Thanks for your advice. The references have been revised as the right form.

d.             Table 4 can be transferred to the supporting material section.

Thanks for your advice. Table 4 has been transferred to the supporting material section.

Reviewer 2 Report

Manuscript ID: Molecules-517710

Title: Data fusion infrared signals to identify geographical origin of wild Paris polyphylla varyunnanensis.

The authors report an interesting approach to identify geographical origin of wild Paris polyphylla varyunnanensis employing Fourier transform mid-infrared (MIR) and near-infrared (NIR) spectroscopies coupled to chemometric models based on partial least squares-discriminant analysis (PLS-DA) and random forest (RF) as an analytical method. I recommend the publication of the paper after few minor revisions, that should help to improve the readability of the manuscript for the Molecules audience, as follows:

Title

In the present form, I think that the Title is not complete, and it presents a mistake related to use the term “data fusion infrared signal”. Please replace by “Chemometric models coupled to Fourier transform mid-infrared (MIR) and near-infrared (NIR) spectroscopies to identify geographical origin of wild Paris polyphylla varyunnanensis”. The correct example of data fusion could be data collected from liquid chromatography (LC) with dual diode array and fluorescence detection (DAD and FLD), which can be

seen in the following reference: Rocío B. Pellegrino Vidal, Gabriela A. Ibañez, Graciela M. Escandar, Advantages of Data Fusion: First Multivariate Curve Resolution Analysis of Fused Liquid Chromatographic Second-Order Data with Dual Diode Array-Fluorescent Detection, Anal.ytical Chemistry 895 (2017) 3029-3035. Doi: https://doi.org/10.1021/acs.analchem.6b04720.

Abstract

Page 1, lines 19-22 “Feature variables extraction (Principal component analysis) and important variables selection methods (Recursive feature elimination and Boruta) were applied for geographical origin traceability, while the classification ability of models with the former method are better than with the latter”. Please, replace the sentence with “Feature variables extraction (principal component analysis-PCA) and important variables selection models (recursive feature elimination and Boruta) were applied for geographical origin traceability, while the classification ability of models with the former model are better than with the latter”.

Comments to authors about nomenclature concerning analytical technique, analytical method, model, algorithm. The authors interchange frequently the meaning of method, model and algorithms in the manuscript. Please see the following reference: Analytica Chimica Acta 806 (2014) 8-26.  

Thus, some examples of selected mistakes are:

1. Introduction

Page 2, line 51 “classification of P. yunnanensis were widely used chemometrics methods combined with…..”.  Please replace by “classification of P. yunnanensis were widely used chemometric models combined with…….”

2. Results and discussion

Page 4, line 115: “pretreatment methods are applied in Table S1, including parameters of cumulative…”.  Please replace by “pretreatment algorithms are applied in Table S1, including parameters of cumulative…”

Page 4, line 118: “For FT-MIR spectra dataset, the worst classification ability was by FD method, which even…”.  Please replace by “For FT-MIR spectra dataset, the worst classification ability was by FD algorithm, which even…”

Page 4-5, lines 119-123: “But for NIR spectra dataset, SNV pretreatment method was the worst preprocessing method. However, SD was the best preprocessing method both for FT-MIR and NIR that the accuracy even reached 100%. Among all preprocessing methods, the best pretreatment method (SD) for each kind of spectroscopy should be selected and used to establish geographical classification models”. Please replace by “But for NIR spectra dataset, SNV pretreatment algorithm was the worst preprocessing algorithm. However, SD was the best preprocessing algorithm both for FT-MIR and NIR that the accuracy even reached 100%. Among all preprocessing algorithm, the best pretreatment algorithm (SD) for each kind of spectroscopy should be selected and used to establish geographical classification models”.

Please correct these types of mistakes in the manuscript, another example can be found in page 11, line 240 “The results of PLS-DA and RF models basing on RFE, Bo and PCs selection methods”.

The authors are asked to improve the quality of all figures.

Author Response

Dear Reviewer,

Thank you for your letter and reviewers’ comments concerning about our manuscript entitled “Data fusion infrared signals to identify geographical origin of wild Paris polyphylla var. yunnanensis” (ID: molecules-517710). Those comments are valuable and very helpful for revising and improving our paper, as well as the important guiding significance to our researches. We have read comments carefully and have made corrections which we hope to meet with approval. Revised portion are marked in red font in the revised paper. The main corrections and the responds to the reviewer’s comments are displayed below:

Reviewer #2:

Questions:

1. Title

In the present form, I think that the Title is not complete, and it presents a mistake related to use the term “data fusion infrared signal”. Please replace by “Chemometric models coupled to Fourier transform mid-infrared (MIR) and near-infrared (NIR) spectroscopies to identify geographical origin of wild Paris polyphylla var. yunnanensis”. The correct example of data fusion could be data collected from liquid chromatography (LC) with dual diode array and fluorescence detection (DAD and FLD), which can be seen in the following reference: Rocío B. Pellegrino Vidal, Gabriela A. Ibañez, Graciela M. Escandar, Advantages of Data Fusion: First Multivariate Curve Resolution Analysis of Fused Liquid Chromatographic Second-Order Data with Dual Diode Array-Fluorescent Detection, Anal.ytical Chemistry 895 (2017) 3029-3035. Doi: https://doi.org/10.1021/acs.analchem.6b04720.

Thanks for your advice. The title has been revised as “Chemometric models coupled to Fourier transform mid-infrared (MIR) and near-infrared (NIR) spectroscopies to identify geographical origin of wild Paris polyphylla var. yunnanensis”.

2. Abstract

Page 1, lines 19-22 “Feature variables extraction (Principal component analysis) and important variables selection methods (Recursive feature elimination and Boruta) were applied for geographical origin traceability, while the classification ability of models with the former method are better than with the latter”. Please, replace the sentence with “Feature variables extraction (principal component analysis-PCA) and important variables selection models (recursive feature elimination and Boruta) were applied for geographical origin traceability, while the classification ability of models with the former model are better than with the latter”.

Thanks for your advice. This sentence has been revised.

3.Comments to authors about nomenclature concerning analytical technique, analytical method, model, algorithm. The authors interchange frequently the meaning of method, model and algorithms in the manuscript. Please see the following reference: Analytica Chimica Acta 806 (2014) 8-26. 

Thus, some examples of selected mistakes are:

(1). Introduction

Page 2, line 51 “classification of P. yunnanensis were widely used chemometrics methods combined with…..”.  Please replace by “classification of P. yunnanensis were widely used chemometric models combined with…….”

(2). Results and discussion

Page 4, line 115: “pretreatment methods are applied in Table S1, including parameters of cumulative…”.  Please replace by “pretreatment algorithms are applied in Table S1, including parameters of cumulative…”

Page 4, line 118: “For FT-MIR spectra dataset, the worst classification ability was by FD method, which even…”.  Please replace by “For FT-MIR spectra dataset, the worst classification ability was by FD algorithm, which even…”

Page 4-5, lines 119-123: “But for NIR spectra dataset, SNV pretreatment method was the worst preprocessing method. However, SD was the best preprocessing method both for FT-MIR and NIR that the accuracy even reached 100%. Among all preprocessing methods, the best pretreatment method (SD) for each kind of spectroscopy should be selected and used to establish geographical classification models”. Please replace by “But for NIR spectra dataset, SNV pretreatment algorithm was the worst preprocessing algorithm. However, SD was the best preprocessing algorithm both for FT-MIR and NIR that the accuracy even reached 100%. Among all preprocessing algorithm, the best pretreatment algorithm (SD) for each kind of spectroscopy should be selected and used to establish geographical classification models”.

Please correct these types of mistakes in the manuscript, another example can be found in page 11, line 240 “The results of PLS-DA and RF models basing on RFE, Bo and PCs selection methods”.

Thanks for your advice. These types of mistakes in manuscript have been revised as right forms.

4.The authors are asked to improve the quality of all figures.

Thanks for your advice. The quality of all figures has been improved.

Reviewer 3 Report

molecules-517710

This manuscript deals with the identification and classification of geographical origins of wild Paris yunnanensis samples using low-, mid- and high-level data fusion strategies combined with feature extraction and important variables selection methods to fuse the chemical information from FT-MIR and NIR spectroscopies.

The manuscript has been written well by the authors.

I recommend this manuscript to be published in Molecules.

The minor corrections should be introduced:

·         Section 2.1. Spectroscopic analysis should be discussed in more detail.

The analyzed spectra can be divided into five distinct ranges: 3700 - 2000, 1800 - 1500, 1500 - 1200, 1200 - 900, 900 - 650 cm-1. Such a system is characteristic of the main components of plant cells - polysaccharide substances such as cellulose and pectin or lignin.

It should be pointed that in the FTIR spectra of P. yunnanensis samples the broad absorption band in the range 3000-3700 cm-1 corresponds to the stretching vibrations of free hydroxyl groups ν(OH) and the groups involved in intra- and intermolecular hydrogen bonds.

The bands observed in the 2800 - 3000 cm-1 range should be assigned
to the following normal vibrations: nas(CH3) 2920 - 2960 cm-1, nas(CH2) 2900 - 2930 cm-1, ns(CH3) 2880 - 2900 cm-1, ns(CH2) 2850 - 2860 cm-1.

Deconvolution of broad bands in the 1500 - 1800 cm-1, 1200 - 1500 cm-1, 950 - 1200 cm-1 range into Lorentzian components is particularly useful.

For example the component with the maximum intensity occurring at the wavenumber 1743 cm-1 corresponds to the asymmetric stretching νas(COO) vibrations of free carboxyl groups of pectins or/and fatty acids. Band at 1653 cm-1 refers to the asymmetric stretching vibrations of carboxyl groups participating in the estric or/and hydrogen bonds. Other component which appears at around 1610 cm-1 corresponds to ns(COO) vibrations of the carboxyl group present in pectin. In this range amide I band is observed too, and band corresponding to ν(C=C) of fatty aids, flavonoids.

Individual bands can be attributed to characteristic vibrations: asymmetric CH3 deformation δas(CH3) and CH2 scissoring vibration δs(CH2) at 1456, in-plane bending δ(OH···O) at 1414 cm-1, CH3 symmetric deformation   δs(CH3) at 1370 cm-1 (Socrates, 2001).

·         Editorial mistakes should be corrected e.g. section 2.3., line156 "2.3. variables datasets...".

·         Figures 1, 4, S7 and S9 : "Wavenumber (cm-1)" should be written.

·        Figures: fonts should be increased; the signature/caption should be placed under the picture.  

·        Fonts of captions of figures should have the same style.

·        Fonts of captions of tables should have the same style.

·         All references should be unified according to Guide for Authors (Abbreviated Journal Name, page range).

Socrates, G., Infrared and Raman Characteristic group Frequencies, third ed., J. Wiley & Sons, LTD, Chichester, New York, Weinheim, Toronto, Brisbane, Singapore, 2001.

Author Response

Dear Reviewer,

Thank you for your letter and reviewers’ comments concerning about our manuscript entitled “Data fusion infrared signals to identify geographical origin of wild Paris polyphylla var. yunnanensis” (ID: molecules-517710). Those comments are valuable and very helpful for revising and improving our paper, as well as the important guiding significance to our researches. We have read comments carefully and have made corrections which we hope to meet with approval. Revised portion are marked in red font in the revised paper. The main corrections and the responds to the reviewer’s comments are displayed below:

Reviewer #3:

The minor corrections should be introduced:

Questions:

1.Section 2.1. Spectroscopic analysis should be discussed in more detail.

The analyzed spectra can be divided into five distinct ranges: 3700 - 2000, 1800 - 1500, 1500 - 1200, 1200 - 900, 900 - 650 cm-1. Such a system is characteristic of the main components of plant cells - polysaccharide substances such as cellulose and pectin or lignin.

It should be pointed that in the FTIR spectra of P. yunnanensis samples the broad absorption band in the range 3000-3700 cm-1 corresponds to the stretching vibrations of free hydroxyl groups ν(OH) and the groups involved in intra- and intermolecular hydrogen bonds.

The bands observed in the 2800 - 3000 cm-1 range should be assigned to the following normal vibrations: nas(CH3) 2920 - 2960 cm-1, nas(CH2) 2900 - 2930 cm-1, ns(CH3) 2880 - 2900 cm-1, ns(CH2) 2850 - 2860 cm-1.

Deconvolution of broad bands in the 1500 - 1800 cm-1, 1200 - 1500 cm-1, 950 - 1200 cm-1 range into Lorentzian components is particularly useful.

For example the component with the maximum intensity occurring at the wavenumber 1743 cm-1 corresponds to the asymmetric stretching νas(COO) vibrations of free carboxyl groups of pectins or/and fatty acids. Band at 1653 cm-1 refers to the asymmetric stretching vibrations of carboxyl groups participating in the estric or/and hydrogen bonds. Other component which appears at around 1610 cm-1 corresponds to ns(COO) vibrations of the carboxyl group present in pectin. In this range amide I band is observed too, and band corresponding to ν(C=C) of fatty aids, flavonoids.

Individual bands can be attributed to characteristic vibrations: asymmetric CH3 deformation δas(CH3) and CH2 scissoring vibration δs(CH2) at 1456, in-plane bending δ(OH···O) at 1414 cm-1, CH3 symmetric deformation   δs(CH3) at 1370 cm-1 (Socrates, 2001).

Thanks for your advice. The expression of “For all above these useful regions, FT-MIR spectra can be divided into five distinct ranges, including 3700 to 2000, 1800 to 1500, 1500 to 1200, 1200 to 900 and 900 to 650 cm-1. In the region of 3700 to 2000 cm-1, the broad absorption band in the range 3700-3000 cm-1 corresponds to the stretching vibrations of free hydroxyl groups ν(OH) and the groups involved in intra- and intermolecular hydrogen bonds. Absorption at the range of 3000 to 2800 cm-1 was assigned to normal vibration mode such as the CH3 asymmetric normal vibration mode at 2960 - 2920 cm-1, CH2 asymmetric normal vibration mode at 2930 to 2900 cm-1, CH3 symmetric normal vibration mode at 2900 to 2880 cm-1, CH2 symmetric normal vibration mode at 2860 to 2850 cm-1. Additionally, the region two to the region four were useful to make deconvolution the bands into Lorentzian components, and the detail information can be observed in Table 1. Moreover, amide I band is observed in the region of 1800 to 1500 cm-1 too, which corresponding to C=C stretching mode of fatty acids and flavonoids.” has been added and the Table 1 has been revised. Besides, the reference has been added.

2.Editorial mistakes should be corrected e.g. section 2.3., line156 "2.3. variables datasets...".

Thanks for your advice. The expression of “variables datasets selected for mid-level data fusion” has been revised as “Important variables datasets selected for mid-level data fusion”.

3.Figures 1, 4, S7 and S9 : "Wavenumber (cm-1)" should be written.

Thanks for your advice. The “Wavenumber (cm-1)” have been revised in figure 1, 4, S7 and S9.

4.Figures: fonts should be increased; the signature/caption should be placed under the picture. 

Thanks for your advice. The fronts of all figures in manuscript have been increased, and the figure captions in manuscript have been placed under the pictures.

5.Fonts of captions of figures should have the same style.

Thanks for your advice. The Fonts of captions of figures in manuscript have been revised as the same style.

6.Fonts of captions of tables should have the same style.

Thanks for your advice. The Fonts of captions of tables in manuscript have been revised as the same style.

7.All references should be unified according to Guide for Authors (Abbreviated Journal Name, page range).

Thanks for your advice. All references in manuscript have been unified.

Socrates, G., Infrared and Raman Characteristic group Frequencies, third ed., J. Wiley & Sons, LTD, Chichester, New York, Weinheim, Toronto, Brisbane, Singapore, 2001.

Reviewer 4 Report

The manuscript deals with a development of a chemometric approach for characterizing of geographical origin of wild Paris polyphylla var. yunnanensis.

Information extracted from FT-MIR and NIR data fusion coupled with different multivariate methodologies allowed building of models able to effectively identify the origin of P. yunnanensis.

The analytical approaches used in the research work are well known and wide used, as reported in literature. Nevertheless, the analytical strategy applied on the fusion approach is interesting and the results appear to be satisfactory and well described.

However, the manuscript could be improved in some critical points.

Comment:

lines 362-364. ATR accessories usually work by recording reflectance signals.

lines 382-384. The value 0 and 1 are used in PLS modelling to assign a class, but in class prediction, what have criteria been adopted? 0.5 to 1.0 or else?

In my opinion, the content of the paper might be publishable but it needs a minor revision.

Author Response

Dear Reviewer,

Thank you for your letter and reviewers’ comments concerning about our manuscript entitled “Data fusion infrared signals to identify geographical origin of wild Paris polyphylla var. yunnanensis” (ID: molecules-517710). Those comments are valuable and very helpful for revising and improving our paper, as well as the important guiding significance to our researches. We have read comments carefully and have made corrections which we hope to meet with approval. Revised portion are marked in red font in the revised paper. The main corrections and the responds to the reviewer’s comments are displayed below:

Reviewer #4:

Questions:

1.     lines 362-364. ATR accessories usually work by recording reflectance signals.

Thanks for your advice. I am sorry that no modification for the initial measurement type due to our negligence, which caused we have to convert the data.

2.     lines 382-384. The value 0 and 1 are used in PLS modelling to assign a class, but in class prediction, what have criteria been adopted? 0.5 to 1.0 or else?

Thanks for your advice. For each sample, the probability of being assigned to each class can be obtained, and the category with the highest probability was seen as the category of this sample. Besides, the explanation has been added in manuscript.

Round 2

Reviewer 1 Report

I am not certain that this manuscript is now closer to the commonly accepted standards.
Authors ignored several suggestions and in my opinion considerable alternations are necessary before it is accepted for publication.  

 First of all, it is not easy to understand vast fragments of the manuscript. Please correct the English.
-    Assignment proposed in Table 1 neglects water bands. A number of features in the presented spectra, including those with maxima at 5169, 3382, 3334 and 1653cm-1, can be attributed to HB water molecule vibrations.
-    C-H stretching vibration definitely can not be found in the fingerprint region.
-    Authors use terms and expressions which are not commonly accepted or have other meaning than intended (e.g. peak assignment instead of band assignment, estric, smoothing region, advancing ATR correction, just to mention a few examples).
-    Minor mistakes can be found in the reference section.
-    Please think about reformulation of the title.

Author Response

Dear Reviewer,

Thank you for your letter and reviewers’ comments concerning about our manuscript entitled “Data fusion infrared signals to identify geographical origin of wild Paris polyphylla var. yunnanensis” (ID: molecules-517710). Those comments are valuable and very helpful for revising and improving our paper, as well as the important guiding significance to our researches. We have read comments carefully and have made corrections which we hope to meet with approval. Revised portion are marked in red font in the revised paper. The main corrections and the responds to the reviewer’s comments are displayed below:

Reviewer #1:

Questions:

1.First of all, it is not easy to understand vast fragments of the manuscript. Please correct the English.

Thanks for your advice. The style, spelling and grammar of our manuscript have been revised by trying our best.

The expression of “difficulty” in line 59 has been revised as “difficult”.

The expression of “were” in line 181 has been revised as “was”.

The expression of “variables” in line 183 has been revised as “variable”.

The expression of “was remained” in line 190 has been revised as “remained”.

The bold form of “The important variables of Boruta algorithm and RFE algorithm of random forest models based on the total” in lines 243-244 has been revised as normal.

The expression of “are” in line 316 has been revised as “is”.

The expression of “truly class” in line 322 has been revised as “true class”.

The expression of “results” in lines 333 and 336 have been revised as “result”.

The expression of “high frequency” in lines 376 has been revised as “high-frequency”.

The expression of “that based on” in line 460 has been revised as “based on”.

The expression of “Acknowledgements” in line 490 has been revised as “Acknowledgments”.

The expression in lines 136-140 has been revised as “But for NIR spectra dataset, SNV pretreatment algorithm was the worst preprocessing algorithm. However, SD was the best preprocessing algorithm both for FT-MIR and NIR that the accuracy even reached 100%. Among all preprocessing algorithm, the best pretreatment algorithm (SD) for each kind of spectroscopy should be selected and used to establish geographical classification models.”.

The lines 180-181 has been revised as “For the two RF models, the initial ntree were defined both to be 2000, and the initial mtry were set as 33 for FT-MIR dataset and 39 for NIR dataset, respectively.”.

2. Assignment proposed in Table 1 neglects water bands. A number of features in the presented spectra, including those with maxima at 5169, 3382, 3334 and 41325px-1, can be attributed to HB water molecule vibrations.

Thanks for your advice. The expressions about hydrogen bond water molecule vibrations have been added in Table 1. And the expression of “Besides, the peaks at 5169, 3382, 3334 and 1653 cm-1 were considered also may corresponding to hydrogen bond stretching and scissoring vibration mode attributed to water molecules [1].” has been added.

[1] Socrates, G., Infrared and Raman Characteristic group Frequencies, third ed., J. Wiley & Sons, LTD, Chichester, New York, Weinheim, Toronto, Brisbane, Singapore, 2001.

Table 1. Peak assignments on the FT-MIR and NIR spectra of wild P. yunnanensis.

Spectral type

Wavenumber   (cm-1)

Base group   and vibration mode

Contribution

NIR

8347

CH, NH and OH stretching   vibration mode

CH2,   Saccharides and glycosides

7256

CH stretching and deformation   vibration mode

CH2

6950

CH, NH and OH stretching   vibration mode

CH2,   Saccharides and glycosides

6324

CH, NH and OH stretching   vibration mode

CH2,   Saccharides and glycosides

5686

CH, NH and OH stretching   vibration mode

CH2,   Saccharides and glycosides

5169

CH, NH and OH and   hydrogen bond stretching vibration mode

CH2,   Saccharides, glycosides and water molecule

FT-MIR

3382

OH asymmetric and hydrogen bond stretching vibration mode

Saccharides,   glycosides and water molecule

3334

OH asymmetric and hydrogen bond stretching vibration mode

Saccharides,   glycosides and water molecule

2930

CH asymmetric stretching vibration   mode

CH2 and CH3

1743

CO stretching vibration mode

free carboxyl   groups of pectins or/and fatty acids

1653

asymmetric   stretching vibrations of carboxyl groups participating in the hydrogen bonds and   hydrogen bond scissoring vibration mode

flavonoids,   saccharides, steroid saponin and water molecule

1610

COO symmetric   normal vibrations mode

the carboxyl   group present in pectin

1456

CH3   asymmetric deformation and CH2 scissoring vibration

CH2 and   CH3

1414

CH symmetric bending vibration mode   and OHO in-plane bending mode

CH2

1370

CH symmetric deformation vibration   mode

CH3

1242

CO stretching vibration mode

saccharides   and oils

1150

CC and CO stretching and COH bending vibration mode

Saccharides   and glycosides

1078

CC and CO stretching and COH bending vibration mode

Saccharides   and glycosides

1020

CC and CO stretching and COH bending vibration mode

Saccharides   and glycosides

922

Sugar   skeleton vibration mode

Saccharides

3. C-H stretching vibration definitely can not be found in the fingerprint region.

Thanks for your advice. The writing mistake in lines 100-101 of “C─H stretching vibration” has been revised as “C─C stretching vibration”.

4.Authors use terms and expressions which are not commonly accepted or have other meaning than intended (e.g. peak assignment instead of band assignment, estric, smoothing region, advancing ATR correction, just to mention a few examples).

Thanks for your advice. The wrong terms and expression in this manuscript have been revised as follows.

The expression of lines 95-96 has been revised as “The 4000 to 3700 cm-1 and 2620 to 1800 cm-1 absorptions of FT-MIR spectral baseline area and diamond crystal spectral region respectively, which areas provide invalid spectral information for this study”.

The expression of lines 106-110 has been revised as “Absorption at the peaks of 2928 and 2852 cm-1 were assigned to normal vibration mode such as the CH3 asymmetric normal vibration mode at 2960 to 2920 cm-1, CH2 asymmetric normal vibration mode at 2930 to 2900 cm-1, CH3 symmetric normal vibration mode at 2900 - 2880 cm-1, CH2 symmetric normal vibration mode at 2860 to 2850 cm-1”.

The expression of lines 115-116 has been revised as “The bands in the region of 9000 to 4500 cm-1 be associated with the first or second overtones”.

The expression in Table 1 of “asymmetric stretching vibrations of carboxyl groups participating in the esterification or/and hydrogen bonds” has been revised as “asymmetric stretching vibrations of carboxyl groups participating in the hydrogen bonds”.

The expression of “smoothing region” in line 161 has been revised as “smooth region”.

The expression of “advancing ATR correction” in lines 374-375 has been revised as “advanced ATR correction”.

5.Minor mistakes can be found in the reference section.

Thanks for your advice. The form of volumes of references has been revised to italic. Besides, the “Simth” in line 509 has been revised as “Smith”, and the “22(1)” in line 578 has been revised as “22”.

6.Please think about reformulation of the title.

The conclusion has been obtained that the definition of infrared is not clear, and the concept of data fusion is inappropriate according to the opinion of another reviewer. The title revised as “Data fusion of Fourier transform mid-infrared (MIR) and near-infrared (NIR) spectroscopies to identify geographical origin of wild Paris polyphylla var. yunnanensis” according to our careful consideration.